A selective approach to stemming for minimizing the risk of failure in information retrieval systems

Göksel Gökhan gciplak@eskisehir.edu.tr 1
Arslan Ahmet 1
Dinçer Bekir Taner 2
1 Computer Engineering, Eskişehir Technical University , Eskisehir , Turkey
2 Computer Engineering, Muğla Sıtkı Koçman University , Mugla , Turkey
Jan Naeem
Electronic publication date: 2023 Jan 10
Publication date: 2023
Volume: 9
Electronic Location ID: e1175
Received 2022 Sep 16; Accepted 2022 Nov 9
Copyright: ©2022 Göksel et al.
Copyright year: 2022
Copyright holder: Göksel et al.
License: This is an open access article distributed under the terms of the Creative Commons Attribution License, which permits unrestricted use, distribution, reproduction and adaptation in any medium and for any purpose provided that it is properly attributed. For attribution, the original author(s), title, publication source (PeerJ Computer Science) and either DOI or URL of the article must be cited.
License URL: https://creativecommons.org/licenses/by/4.0/

Keywords: Selective information retrieval, Selective stemming, Robustness

Funding: TÜBİTAK, scientific and technological research projects funding program 114E558 This work is supported by the TÜBİTAK, scientific and technological research projects funding program, under Grant 114E558. The funders had no role in study design, data collection and analysis, decision to publish, or preparation of the manuscript.

==============================
Stemming is supposed to improve the average performance of an information retrieval system, but in practice, past experimental results show that this is not always the case. In this article, we propose a selective approach to stemming that decides whether stemming should be applied or not on a query basis. Our method aims at minimizing the risk of failure caused by stemming in retrieving semantically-related documents. The proposed work mainly contributes to the IR literature by proposing an application of selective stemming and a set of new features that derived from the term frequency distributions of the systems in selection. The method based on the approach leverages both some of the query performance predictors and the derived features and a machine learning technique. It is comprehensively evaluated using three rule-based stemmers and eight query sets corresponding to four document collections from the standard TREC and NTCIR datasets. The document collections, except for one, include Web documents ranging from 25 million to 733 million. The results of the experiments show that the method is capable of making accurate selections that increase the robustness of the system and minimize the risk of failure (i.e., per query performance losses) across queries. The results also show that the method attains a systematically higher average retrieval performance than the single systems for most query sets.

Introduction

Stemming serves two purposes in the context of Information Retrieval (IR): (i) reducing the index size and (ii) recall enhancement in the document lists retrieved for queries. Mapping actual query terms to their base forms reduces the amount of unique terms in any given collection and as a result it proportionally reduces the size of the index. In this respect, stemming serves well and there is no counter factual issue reported in the IR literature. On the other hand, this process also increases the relative frequency of terms within a document by unifying morphological variants of the terms and hence it is supposed that it increases the level of recall in the document lists retrieved by IR systems to any given query. However, this unification process may unify morphologically related but semantically unrelated two terms into a single base form. In such a case, stemming causes those documents which are semantically unrelated to a given query to become, unexpectedly, a part of the document list retrieved for that query. That’s why stemming harms the performance for some queries by reducing the level of recall in the document lists retrieved for the queries.

It has been a long-standing debate whether stemming contributes to the effectiveness of an IR system (Harman, 1991; Krovetz, 1993; Alotaibi & Gupta, 2018). On this account, Harman (1991) shows that stemming does not affect average retrieval performance but rather, it affects both positively and negatively to an almost equal number of queries and hence the average remains the same. However, in contrast, the works of Krovetz (1993) and Hull (1996) provide evidence that counts against those negative empirical findings and relate them to the system and the query sets examined in the work of Harman (1991). In this respect, it can be said that stemming may or may not contribute to the retrieval effectiveness of an IR system depending on the query that the system responds to.

In this paper, to decide whether stemming should be applied to any given query, a selective approach is proposed. The approach is based on k-Nearest neighbors binary classifier and it employs a base set of existing pre-retrieval query performance features from the IR literature, including inverse document frequency (IDF), query scope (He & Ounis, 2004a), and average similarity between collection and query (AvgSCQ) (Zhao, Scholer & Tsegay, 2008). In addition, a set of new features that are derived from the frequency distributions of query terms is introduced in this paper and used for the binary classifier (Section ‘Proposed selective approach to stemming’). The proposed selective approach follows the common IR practice, where BM25 (Robertson & Zaragoza, 2009) is used as the term weighting model and rule-based stemmers.

It is expected that an IR system fulfill every information need of users at an acceptable level of satisfaction. Such a system refers to a robust system that evenly distributes its total (average) effectiveness on every query. If a system applies stemming and its average performance remains unchanged (the results of the work of Harman), it is highly likely that the system increases performance for some queries and decreases performance for some other queries: that is, the system diverges from being robust. Such situations lead to high variation in performance across queries, and hence harms the robustness of IR systems. The proposed selective approach is a remedy to this problem and it is capable of providing robustness in retrieval effectiveness for the IR systems employing stemming (Section ‘Evaluation of the Selective Approach to Stemming’). By alleviating the problem, the performance of an IR system can be improved more than the expected performance of the system in which stemming is naively applied. In this perspective, the paper makes a contribution to the IR literature by considering the importance of the selective application of stemming in IR systems.

In particular, we address the following research questions:

RQ1 To what degree the proposed selective approach is accurate in predicting the queries that stemming should (not) be applied to?

RQ2 Does the proposed selective approach contribute to the robustness in retrieval effectiveness of the IR systems that employ stemming?

In summary, the work presented in this paper contributes to the literature by proposing a selective approach to stemming. The work aims at minimizing the risk of failure in retrieving semantically related documents to the employed stemming algorithm for any given query. We have achieved this aim by accurately predicting whether stemming should be applied to a given query using a machine-learning technique. We have used a set of query performance predictors from the literature and new features for this purpose. Also, the risk-sensitive analysis shows that the proposed work increases the robustness of an IR system by minimizing the performance fluctuation across queries caused by a failure in applying stemming to the IR system. According to the experimental results, the proposed selective method not only increases the overall retrieval performance in most cases but also contributes to the robustness of the IR system together. The selective method validates its contribution to the robustness of the IR system according to the performed experiments on diverse sets of queries and generalizes to the rule-based stemming algorithms. Considering the contribution, aim, and used method together, the proposed paper includes pioneering work to our best knowledge.

The rest of the article is organized as follows. The motivation of this work is provided in Section ‘Motivation’. The section ‘Related Works’ reviews the related works about selective IR, stemming, and selective stemming. Differences between the proposed work from prior work are also presented in the section. Details of the proposed work are given in Section ‘Proposed Selective Approach to Stemming’. The experimental evaluations are performed on a wide range of standard set of queries from TREC and NTCIR (Section ‘Experimental Setup’). The used document collections of the corresponding query sets are based on Web corpora that includes ClueWeb09-B, ClueWeb12-B13, GOV2, and, also Wall Street Journal (WSJ) newspaper articles in TIPSTER collection is used. On the query sets, the evaluation results (Section ‘Evaluation of the Selective Approach to Stemming’) show that our selective method is on average more effective and robust than the considered single systems that are participated in selection procedure for most of the query sets and stemming algorithms. Implications of this work are discussed in Section ‘Implications’ from theoretical and practical perspective.

Motivation

The factors of per query performance variability between retrieval strategies are extensively studied at the reliable information access workshop for robust retrieval (Harman & Buckley, 2004; Buckley, 2004). To identify those factors, Buckley (2004) collates the reasons why IR systems fail for individual queries into ten categories of which stemming is a category identified as general technical failures. On this account, Harman (1991) says that queries are affected by stemming positively or negatively:

“Although individual queries were affected by stemming, the number of queries with improved performance tended to equal the number with poorer performance, thereby resulting in little overall change for the entire test collection.”

For instance, an IR system in which stemming is applied has produced a score of 0.1568 for the query poker tournaments (TREC Web Track 2009 QueryID:17), while the system without stemming has produced a score of 0.4132 for that query. However, applying stemming to the query mothers day songs (TREC Web Track 2011 QueryID:132) has provided higher performance than without applying stemming in the IR system.

On the other hand, Buckley (2004, 2009) says that it may be more important to determine what strategies should be applied to which queries, rather than developing new IR strategies. In this context, stemming can be applied in a selective, binary classification manner, in such a way to improve the robustness of an IR system, so as to alleviate the performance variability across queries. Thus, a selective approach to stemming can avoid the issue mentioned in the work of Harman (1991) to a certain degree.

Figure 1 shows, for 100 queries from the NTCIR-13 WWW-1 track (Luo et al., 2017), that the within query difference in performance score between the run BM25 with stemming (STEM) and the run BM25 without stemming (NOSTEM), where stemming algorithms is Krovetz stemmer (Krovetz, 1993). The x-axis of the figure shows the query number, and the y-axis is the score difference between the two runs for the corresponding query. On the left side of the figure, the queries within which NOSTEM has higher scores than STEM are shown (37%), and on right side the queries within which STEM > NOSTEM (39%). The queries on which STEM = NOSTEM are shown in the middle of the figure (24%). On average, the BM25 run with stemming, STEM has an nDCG@20 score of 0.3709 and the run NOSTEM 0.3755. Here, the difference between two average nDCG@20 scores can statistically be attributed to chance fluctuation (p-value = 0.70 for the paired t-test). Considering the observed difference in average performance scores, it would appear that, even for a recent official evaluation effort, NTCIR-13 WWW-1 track, queries are affected by stemming positively or negatively as mentioned in the work of Harman (1991). This suggests that Harman’s argument still keeps its strength.

Figure 1 The NTCIR-13 WWW-1 track for 100 queries, that the per query performance score difference between BM25 with stemming (STEM) and BM25 without stemming (NOSTEM), where stemming algorithms is Krovetz stemmer.

In this context, it can be said that, following the Buckley’s argument, determining in advance which queries should be stemmed, that is, selective stemming, would be a solution. Note that here, a perfect selective approach to stemming, an oracle method could achieve an nDCG@20 score of 0.4041 on average, which is statistically better in retrieval performance than both STEM (p-value = 0.0004) and NOSTEM (p-value = 0.00002). In conclusion, the aim of this study is to minimize the risk of failure by improving the robustness of an IR system by automatically deciding whether stemming should be applied to a given query.

Related Works

Selective IR is addressed by applying a particular retrieval strategy to the given query. These strategies are used in the pre-retrieval to post-retrieval phases of the IR system, depending on the applied technique. This section primarily presents studies on selective IR applied in phases of an IR system. Then, we review the stemming algorithms in the literature by grouping them into rule-based and corpus-based. Selective approaches to stemming in IR systems is reviewed in the next section. The last section gives the differences from prior work.

Selective IR

Selective approaches can be applied to any phase of an IR system, as each phase usually encompasses different techniques that can be selected. Indexing a document collection is the first phase of the IR system in which a selective approach has the potential to be applied. Large-scale document collections are partitioned into several topically homogeneous groups named shards. Searching is only executed on a few shards that are postulated to involve relevant documents for a given query, which is called selective search. This strategy aims to reduce the retrieval cost for the query by using only a small piece of collections and preserving retrieval effectiveness as possible as that of an exhaustive search. In this respect, resource selection algorithms and techniques to select shards are proposed (Kulkarni & Callan, 2015; Kim et al., 2016; Kim et al., 2017). Each sub-component of IR systems affects the retrieval performance on a query basis. Considering the studies, it is seen that this research area is rich in literature.

As part of an IR system, term weighting models affect per query performance variance, and queries do not benefit equally from term weighting models. Therefore, selective approaches to term weighting model selection (He & Ounis, 2003; He & Ounis, 2004b; Arslan & Dinçer, 2019) are presented to alleviate performance degradation across term weighting models.

For another part of an IR system, the selective approach to Learning-to-Rank (LTR), which reranks retrieved documents with a learned model, is presented (Peng, Macdonald & Ounis, 2010; Balasubramanian & Allan, 2010; Ghanbari & Shakery, 2019). As in the term weighting models, different queries take advantage of each ranking function differently and selective methods are studied to decide appropriate function on a per query basis. In addition, the researchers in the work (Tonellotto, Macdonald & Ounis, 2013) proposed a selective pruning framework. Their work determines if the result list of the query should be pruned aggressively.

Query expansion techniques append new terms to the query to increase the recall of an IR system by matching more documents. However, it is not always the case that any query makes use of the appended terms in terms of retrieval effectiveness. Hence, selective approaches to query expansion are proposed to determine whether it should be applied (Amati, Carpineto & Romano, 2004; Cronen-Townsend, Zhou & Croft, 2004; Hauff et al., 2010) and also to select the terms being added to the query (Cao et al., 2008; Saleh & Pecina, 2019).

Optimization of an IR system configuration that seeks to maximize the performance of the IR system by predicting appropriate techniques is a research area in IR. The studies in this area (Bigot, Dejean & Mothe, 2015; Deveaud et al., 2018; Mothe & Ullah, 2021) aim to predict the most appropriate combination of the IR system components ranging from indexing to document ranking. The common point of these studies is to deal with all parts of the system in its entirety.

Stemming

Stemming is a remedy for vocabulary mismatch, and the application of stemming to an IR system takes place in the pre-retrieval phase. Being an important recall enhancement tool for the preprocessing phase of an IR system has made stemming a rich place in the literature. In this context, stemming methods can be broadly classified into two groups: (i) rule-based stemming methods and (ii) corpus-based stemming methods. Rule-based stemmers such as Lovins stemmer (Lovins, 1968), Krovetz stemmer (Krovetz, 1993), Porter stemmer (Porter, 1997), and Paice/Husk stemmer (Paice, 1990) transform terms to their morphological roots using language-specific rules. Specifying the rules in a particular language needs expertise in that language. In addition, several linguistic resources can be used in developing a rule-based stemmer. Once the rule is created for a language, it can be used in any corpus without additional processing, making it ease of use. For different kind of languages (i.e., Arabic (Al Kharashi & Al Sughaiyer, 2002; Abuata & Al-Omari, 2015), Croatian (Ljubešić, Boras & Kubelka, 2007), Urdu (Gupta, Joshi & Mathur, 2013), Bengali (Sarkar & Bandyopadhyay, 2008; Mahmud et al., 2014), Marathi (Patil & Patil, 2017)), rule-based stemming approaches are presented in literature. Similar to the rule-based method, removing suffixes or prefixes like pluralization handling is a stemming technique. On the other hand, corpus-based stemmers construct the conflation sets, involving the morphological variants of terms, from a given corpus without requiring any linguistic knowledge. Lexicon analysis with string processing, character n-gram based analysis, co-occurrence of words, and context analysis on a given corpus are common techniques in corpus-based stemming. Lexicon analysis based stemmers group the related words in the corpus words. This strategy is usually performed by the operations such as finding suffixes, suffix stripping, string distance, etc., (Oard, Levow & Cabezas, 2001; Goldsmith, 2001; Paik et al., 2011). In the character n-gram based method, adjacent characters in a length of n from the words in a corpus are considered to have less frequency whereas the variants have higher frequencies (McNamee & Mayfield, 2004; Ahmed & Nrnberger, 2009; Pande, Tamta & Dhami, 2018). Also, various studies on corpus-based stemming using co-occurrence analysis and machine learning techniques are presented (Paik, Pal & Parui, 2011; Paik et al., 2013; Brychcn & Konopk, 2015). These methods analyze the co-occurrence or context of the basis form of the words in a corpus. In this regards, lexical and co-occurrence similarities are usually applied to discover morphologically related words. For instance, the work of Singh & Gupta (2019) also employed suffix pair frequency and graph-based clustering besides lexical and co-occurrence similarity in order to construct the conflation sets. In another work, researchers applied Hidden Markov Model to produce stems of the words (Bölücü & Can, 2019).

A recently proposed method by Singh & Bhowmick (2022) uses neural network to predict co-occurrence similarity between a query term and its potential morphological variant. Potential or candidate variants are initially determined by using lexical similarity of the words in the corpus.

Selective stemming

From the perspective of selective stemming, several studies are presented to decide whether stemming should be applied to an IR system on a per query basis. In the works of Harman (1987, 1991), the selection process was carried out on the basis of two criteria, query length, and term importance. However, the results of the experiments provided no evidence indicating significant improvement in retrieval effectiveness on average. In another work of Harman (1991), the selection was simply based on a threshold: that is, if the query length is shorter than ten terms, stemming is applied; otherwise, no-stemming is applied. Machine learning based selection methods was also proposed in the work of Chin et al. (2010). The method in the work employs Support Vector Regression models built on the query features to select an appropriate text normalization technique among stemming, depluralization, and without any text normalization. On the other hand, advanced optimization algorithms are also used for selective stemming. In this context, a selective approach to stemming based on a genetic algorithm leveraging query performance predictors is used as well (Wood, 2013). In addition to selecting a stemming method, there are different works on selective stemming. Stemming may be applied to an IR system at run time by expanding the query with the morphological variants of its query terms. Thereby, the method that select the morphological variants from the set of candidate variants for a given query is a selective approach to stemming (Croft & Xu, 1995; Peng et al., 2007; Cao, Robertson & Nie, 2008). Particularly, in the work of Tudhope (https://cs.uwaterloo.ca/research/tr/1996/31/cs-96-31.pdf), query expansion was accomplished by means of a filtering strategy that filters out those morphological variants of the query terms that have not the same semantic meaning. This query expansion technique was examined using TREC-4 Ad Hoc task queries, and the results of the experiments show that it improves the accuracy of the stemming algorithm in use. Similarly, in the work of Cao, Robertson & Nie (2008), a query expansion technique that expands the query on the basis of the expected effect of individual variants on retrieval effectiveness was introduced. The study also adopted a further technique that uses the language model to decide the morphological variants of query terms that best fit the query context. The results of the experiments show a significant improvement over the not stemmed system in retrieval effectiveness on GOV2 Web Track and several TREC Ad Hoc track queries. However, the methods in the experiments couldn’t show the same performance as the traditional stemming method that expands the query with all morphological variants of the query terms. The similarity between the query term and its morphological variants may be determined by word embeddings. Word embedding is a vectorial representation of a term so as to measure contextual similarity between given terms. String similarity and contextual similarity are employed to select morphological variants (Basu et al., 2017). The researchers in the work (Roy et al., 2017) used local and global word embeddings to measure contextual similarity within term variants and filter them to create final clusters. Furthermore, in the same line of research, the selection of the term variants based on the language models was also investigated in the work of Peng et al. (2007) utilizing an occurrence analysis of query term variants on the corresponding document collection.

Differences from prior work

The selective stemming method presented in this paper considerably extends previous works (Harman, 1987; Harman, 1991; Chin et al., 2010; Wood, 2013) in which we follow the same research line. One of the limitations of these works is the size of the query sets: the number of tested queries is relatively smaller than the query sets that have been brought to the literature in total recently. Another limitation is the decision methods: even if some of the works build a machine learning model, the features used in the model must be able to discriminate the queries as possible. Features in the works usually could not cover the term frequency distribution yielded by an IR system where stemming is applied. However, stemming does not behave collection and document frequencies of the terms in a corpus similarly since the frequencies of some terms would substantially increase from the others.

Our proposed method is a binary classifier that leverages the not only query performance predictor features but also the derived features from the fluctuation in an increase of the term frequencies for the systems with and without stemming. To our best knowledge, the proposed work in this paper is a pioneering work that quantifies the inter-relation of within query term specificity (Spärck Jones, 1972; Robertson, 2004; Church & Gale, 1999) and derives query features from that information. Those features usually leverage the differences in term specificity for both with and without stemming. This method is evaluated on a diverse set of standard IR test collections using out-of-the-box retrieval configuration. The test collections used in the experiments involve eight query sets related to four comprehensive document collections that broadly include Web documents and also newspaper journals. In addition, we perform a risk-sensitive evaluation to determine the robustness of the proposed selective stemming method. This evaluation indicates that the average performance enhancement by the method is spread to the queries.

Proposed Selective Approach to Stemming

The pioneering works on selective approach to stemming are the works of Harman (1987, 1991). Those works employ only one selection criterion to make the binary decision (i.e., stemming vs. no-stemming). That selection criterion is the length of queries measured in terms of non-common terms, i.e., the number of non-common query terms. Stemming is basically applied to those queries of which the length is less than ten terms. Similarly, in the work of Wood (2013), a selective approach to stemming is also introduced on the basis of a binary decision process. The decision relies on a genetic algorithm utilizing query performance predictors, including inverse document frequency, inverse collection term frequency, average collection query scope, etc. However, the results of the experiments presented in the mentioned works show that the introduced selective approaches provide little or no improvement on the average retrieval effectiveness of the system with stemming. The proposed work in this paper is also a binary classifier for a selective approach to stemming, and it is based on a supervised classification technique known as the k-Nearest neighbor classifier. The classifier employs a set of query performance predictors introduced in the IR literature and a new set of features derived from differences in the term frequencies of the systems participating in the selection.

The frequency distribution of a query term on a given document collection would generally change depending on whether or not stemming is applied to that term. In this study, we assume that the difference in term frequency distribution obtained after applying stemming to a query term can be used as a criterion to decide whether stemming should be applied or not. Therefore, in our study, the features presented in the literature and produced based on our assumptions are used. The query performance predictors that are borrowed from the literature are of pre-retrieval type (Carmel & Yom-Tov, 2010), including minimum IDF ratio over maximum IDF ratio (gamma), query scope (omega) (He & Ounis, 2004a), maximum IDF and AvgSCQ (Zhao, Scholer & Tsegay, 2008). In addition to those four features, we introduce the following six features derived from the frequency distributions of query terms, given as follows:

• AvgIncDF: Stemming increases the document frequency (DF) of a particular term by unifying the posting lists of its morphological variants in IR systems. The increase may not be the same degree for all terms in the query. This feature basically measures the average increase in document frequency for a given query, as in Eq. (1). (1) 1|q|∑i=1|q|DFiStem−DFiNoStemDFiNoStem.

In the equation |q| is the number of query terms in the given query.

• MaxWeightedIncDF: Inverse document frequency measures the term specificity of a term in a document collection (Spärck Jones, 1972; Robertson, 2004). It can be supposed that an importance value calculated by multiplying the increase in the document frequency and term specificity would be an indicator to measure the retrieval effectiveness of applying stemming to the query terms. The maximum importance value among the calculated values for each query term is used as a feature, as given in Eq. (2). (2) MaxIdf1NoStem×DF1Stem−DF1NoStemDF1NoStem,…,IdfnNoStem×DFnStem−DFnNoStemDFnNoStem.

In the equation, n is the number of terms in the query.

• CorrIctfRank: After stemming is applied, in a query, term specificity such as inverse document frequency (Idf) or inverse collection term frequency (Ictf) for a given query term may relatively change according to the other terms in that query. We apply this observation as a feature (it takes 0 or 1) measuring the correlation between the ranks of the term positions of the query terms. Two lists including term positions are constructed for both stemming and no-stemming and they are sorted by term specificity (i.e., Ictf). Here, we use the Spearman rank correlation and we assume that they are correlated if the correlation coefficient value is greater than 0.7.

• Mst-Lst-Change: After applying stemming, the least specific or the most specific term in a given query may change. Those changes are used as a feature. If a change occurs on the least specific or the most specific term, it takes 1; otherwise, 0.

• Chi2-DF-TF: When stemming is applied to the given query, document frequency and collection term frequency of the query terms may be affected more than other terms. We use p-value of the Pearson’s chi-square goodness-of-fit test to see this effect and employ it as a feature. First, we construct two lists involving document and collection term frequencies of the given query terms for both stemming and no-stemming, preserving the term positions. Since the chi-square goodness-of-fit test requires discrete values, a binning strategy, the Freedman-Diaconis rule, is employed to those frequencies to obtain a finite number of bins and groups of the terms. Thus, p-value of the chi-square goodness-of-fit test is obtained with using two lists and employed as a feature.

• ModifiedSCS: This feature is modified version of the simplified query clarity score proposed in the literature (Cronen-Townsend, Zhou & Croft, 2002; He & Ounis, 2004a). (3) ∑QPmlw|Q⋅lnPmlw|QPcollstemw.

In Eq. (3), Pml(w|Q) is the maximum likelihood of the query model as in the proposed definition in the literature. It is given by qtf/ql, where qtf is the frequency of a query term in the query and ql is the query length. Pcoll(stem(w)) is the collection model. Whereas it is defined by Pcoll(w), which is given by tfcoll/tokencoll, in the literature, it is modified by tfcoll/tfcoll(stem(w)), where tfcoll, tokencoll, and tfcoll(stem(w)) are the collection term frequency, the number of total terms in the collection, and the collection term frequency of a stemmed query term respectively. The value of tfcoll(stem(w)) can be calculated by summing collection term frequencies of the terms in the conflation set of the query term at running time.

The labels used in the classifier are derived from retrieval performance scores of both no-stemming (NoStem) and used stemming algorithm in selection. The labeling process using the retrieval performance scores of the queries proceeds as follows: If the retrieval performance score of the query in stemming is higher than the score in no-stemming, the label of the query is assigned as “1” for stemming; otherwise, “0” for NoStem. Furthermore, the queries with the same retrieval performance score for both no-stemming and stemming, tie queries, are discarded in the training phase. The remaining queries are used to build the training model. Consequently, the proposed classifier decides whether or not stemming should be applied for a given query.

Experimental Setup

This section details the setup of the search engine tool and document collections in use. We have used an open-source search engine tool to perform retrieval experiments (Section ‘Experimental system’). In addition, the experiments are carried out using large-scale document collections and their corresponding query sets (Section ‘Benchmark collections’). The proposed selective approach is evaluated with the normalized Discounted Cumulative Gain (nDCG) (Järvelin & Kekäläinen, 2000; Järvelin & Kekäläinen, 2002) for the top 20 documents in the result list. Since the user naturally expects highly relevant documents to be at the top of the result list, especially for the Web collections, the evaluation metric meets this expectation by assigning more scores to the top-ranked relevant documents during the evaluation process. The ‘Baselines’ section describes state-of-the-art baseline methods compared and evaluated in this work.

Experimental system

The IR community addresses the reproducibility of IR experiments and encourages the authors to conduct their experiments following the adopted standards as much as possible (Arguello et al., 2016; Lin et al., 2016; Voorhees, Rajput & Soboroff, 2016). In this respect, details of the experimental system are given for reproducibility of the performed experiments. Source codes of the experiments are made publicly available on GitHub repositories (https://bitbucket.org/gokhanc/lucene-clueweb-retrieval/src/master/) (https://github.com/gokhanc90/matlabIRexperiments).

We use Apache Lucene (Białecki, Muir & Ingersoll, 2012), an open-source search engine platform that is developed for commercial purposes, to perform IR experiments in this study. Although widespread use in the industry is also gradually accelerating its use in academic research (Azzopardi et al., 2017). Indexing HyperText Markup Language (HTML) documents with Lucene, the documents are stripped from their HTML tags using jsoup (https://jsoup.org/) library so as to get plain text blocks of a given HTML document. The final text block to be indexed is obtained by combining the title and body text blocks extracted by the jsoup library into one unstructured text block. The combined text block is processed with StandardTokinezer and LowerCaseFilter of Apache Lucene without employing stemming and stopword removal in the indexing process.

Instead of keeping indexes separately for each stemming algorithm in this study, the query time stemming approach (Peng et al., 2007; Cao, Robertson & Nie, 2008) is utilized by means of applying SynonymGraphFilterFactory of Apache Lucene to the corresponding documents and the given queries. To perform this functionality, query time stemming, the search tool requires a set of morphological variants of the words to be stemmed. For this purpose, morphological variants of each query term, generated by the concerned stemming algorithm, are provided to the system. As a result, the IR system retrieves the documents for a given query according to a particular stemming algorithm over a single index in Apache Lucene. In addition to the stemming procedure mentioned, BM25 term weighting model in Terrier software (http://terrier.org/) is adapted to Lucene (https://lucene.apache.org/) version 7.7.0 platform.

Benchmark collections

The ClueWeb09 collection includes about 1 billion Web documents collected between January and February 2009. The Category B subset of the collection includes about 50 million English Web pages. This subset has been used in TREC Web Tracks ran through 2009 to 2012 (CW09B) and the Million Query Track 2009 (MQ09). The ClueWeb12 collection includes about 733 million English Web documents collected between February 10, 2012 and May 10, 2012. Uniformly extracted 7% sample of the collection is named Category B13. This collection, ClueWeb12-B13, has been used in TREC Web and Tasks Tracks 2013-2016 (CW12B) and the NTCIR (Luo et al., 2017; Mao et al., 2019) We Want Web Tracks 13&14&15 (NTCIR). The GOV2 (Clarke, Craswell & Soboroff, 2004) collection involves about 25 million Web documents from the .gov domain. TREC Terabyte Tracks 2004-2006 (GOV2), Million Query Tracks 2007 (MQ07), and 2008 (MQ08) have been conducted using GOV2. Wall Street Journal collection contains about 173 thousand newspaper articles in the TIPSTER collection of disk 1 & 2. This collection has been used in TREC 1-2-3 Ad Hoc Tasks. The experimental evaluations are carried out on a wide range of standard TREC datasets and their corresponding tracks. The number of queries for each track is summarized in Table 1.

Table 1 Description and number of queries for Tracks according to document collections in the experiments.

Collection	Track	Label in experiment	Number of queries	
WSJ	Ad Hoc 1,2 & 3	WSJ	150	
GOV2	Terabyte 2004, 2005 & 2006	GOV2	149	
	Million Query 2007	MQ07	1,524	
	Million Query 2008	MQ08	564	
CW09B	Million Query 2009	MQ09	562	
	Web 2009, 2010, 2011 & 2012	CW09B	197	
CW12B	Web 2013 & 2014	CW12B	185	
	Tasks 2015 & 2016	
	We Want Web 13 & 14	NTCIR	180	

Baselines

The proposed method decides whether stemming should be applied or not on a query basis. Thereby, two systems participating in this selection are our baselines. NoStem refers to the system in which any stemming method is not applied, and it is the mandatory system involved in the selection. Another one is the system in which one of the KStem (Krovetz, 1993), Porter (Porter, 1980), and Lovins (Lovins, 1968) stemming algorithms is applied. Hereby, we have performed the experiments by the combinations of NoStem and stemming algorithms: NoStem-KStem, NoStem-Porter, and NoStem-Lovins.

Evaluation of the Selective Approach to Stemming

The proposed selective approach to stemming is evaluated by employing common IR practice. To accomplish the practice, BM25 term weighting model (Robertson & Zaragoza, 2009), which is an out-of-the-box option in many IR works, is used in the evaluation of the retrieval effectiveness. Furthermore, the stop-word removal process is not utilized during indexing the documents. The proposed selective approach uses k-Nearest neighbors classification algorithm to decide whether stemming should be applied, where k is chosen as 11 and Minkowski distance with exponent value set to 3 is used to find the closest neighbors. Leave-one-out cross-validation method is employed to evaluate the classifier so that each query is used as a test query during the evaluation process (Arlot & Celisse, 2010).

Retrieval results

Table 2 lists the average nDCG@20 scores of the systems for the query set collections. The proposed method is named by appending the prefix Sel keyword to its baseline stemming algorithm. The highest scores are indicated in boldface. The highlighted scores are the values that NoStem has the best score against the scores of selective method and the corresponding stemmer. The collections of query sets and the number of queries in those sets are given in the first and second columns respectively. We have discarded the queries where at least one of its terms is not found in the document collection. For example, the term tetacycline (TREC 2008 Million Query Track Query ID:16625) does not occur in the GOV2 dataset. The remaining columns show the performance scores of the systems and selective method. B refers that the performance difference between the proposed method and the baseline stemmer is statistically significant with a p-value o f < 0.1. Similarly, N denotes that the performance difference between the proposed method and NoStem is statistically significant with the same p-value.

Table 2 The table presents the nDCG@20 scores of the systems for the query set collections.

The highest scores are indicated in boldface. The italicized scores are the values that NoStem has the best score against the scores of selective method and the corresponding stemmer.

Collection	# Queries	NoStem	KStem	SelKStem	NoStem	Lovins	SelLovins	NoStem	Porter	SelPorter	
CW09B	197	0.1606	0.1523	0.1640 N,B	0.1606	0.1460	0.1634 B	0.1606	0.1497	0.1563	
CW12B	185	0.0781	0.0858	0.0872 N	0.0781	0.0807	0.0849 N	0.0781	0.0872	0.0874 N	
NTCIR	178	0.2946	0.2956	0.3051 N,B	0.2946	0.2681	0.2934 B	0.2946	0.2839	0.2972 B	
GOV2	149	0.3486	0.3697	0.3707 N	0.3486	0.3670	0.3611	0.3486	0.3839	0.3885 N	
WSJ	148	0.3313	0.3661	0.3576N	0.3313	0.3485	0.3417N	0.3313	0.3684	0.3657N	
MQ07	1520	0.2253	0.2313	0.2321 N	0.2253	0.2227	0.2264	0.2253	0.2366	0.2315N	
MQ08	562	0.2623	0.2607	0.2631	0.2623	0.2431	0.2586 B	0.2623	0.2631	0.2609	
MQ09	562	0.2508	0.2543	0.2546	0.2508	0.2433	0.2479	0.2508	0.2552	0.2549	

NoStem has the highest scores for NTCIR, MQ08, and MQ09 collections in Lovins, and for CW09B collection in Porter. The proposed method with KStem produces the highest scores on average for each corresponding collection of query sets except WSJ. Furthermore, it produces a statistically significant improvement in average performance over NoStem for the collections CW09B, CW12B, NTCIR, GOV2, WSJ, MQ07, and significant over KStem for the CW09B and NTCIR collections. However, the proposed method yields higher performance scores than its baseline stemmer (Lovins) except for GOV2 and WSJ collections. The improvements are statistically significantly higher than its baseline stemmer for CW09B, NTCIR, and MQ08 while it is significantly higher than NoStem for CW12B and WSJ. The proposed method with Porter yields the highest scores for CW12B, NTCIR, and GOV2. For the collections CW12B, GOV2, WSJ, and MQ07, the performance of the selective method is statistically significantly higher than NoStem and significantly higher than baseline stemmer for NTCIR.

The most important conclusion from Table 2 is that the selective approach attains a systematically higher average retrieval performance than the single systems for most query sets. Another important result is that the approach does not produce a significantly worse performance score than any single system, even if various collections of query sets are used. On the other hand, selective method could not succeed for WSJ collection. All stemming algorithms produce the highest score for this collection. The reason for this situation can be explained by interpreting Tables 3 and 4. Table 3 lists the average length of the distinct terms in the queries for each query set. It can be easily seen that average length of the terms are close to each other. Table 4 lists the average length of the distinct terms in the corpus of the document collections. WSJ document collection and its corresponding query set Ad Hoc Tracks have close term length on average but other document collections have higher average term length than their corresponding query sets. One of the differences between WSJ and other document collections is that WSJ is a newspaper collection while others include Web documents. Other difference is the number of distinct terms in corpus. WSJ includes quite a few distinct terms compared to other corpus. Therefore, considering the scarcity of documents and the number of distinct terms in the WSJ corpus, it benefits from stemming algorithms alleviating the mismatching problem since more documents can be scored, and more relevant documents can be accessed. Since other corpora contain a large number of documents, it can be considered that this problem will arise less frequently. When stemming is applied, it is usually expected that irrelevant documents will be included in the result list in a collection with so many documents, and this will hurt performance in some queries. This argument can be made as an inference from the highlighted cases in Table 2. For instance, NoStem has the highest scores for the query set CW09B against Porter and for the query sets NTCIR, MQ08, and MQ09 against Lovins.

Table 3 Average character length of the distinct terms in the queries.

Tracks	Sum of distinct term length	# Distinct terms in queries	Average	
Web 2009, 2010, 2011 & 2012	2,457	419	5.86	
Web 2013 & 2014, Tasks 2015 & 2016	2,705	456	5.93	
Terabyte 2004, 2005 & 2006	2,791	413	6.76	
Million Query 2007	18,616	2,824	6.59	
Million Query 2008	9,408	1,475	6.38	
Million Query 2009	7,113	1,140	6.24	
We Want Web 13 & 14	2,458	396	6.21	
Ad Hoc 1,2 & 3	3,159	447	7.07	

Table 4 Average character length of the distinct terms in the corpus of the document collections.

Corpus of document collection	Sum of distinct term length	# Distinct terms in corpus	Average	
WSJ	1,651,354	212,146	7.78	
GOV2	95,309,991	10,440,851	9.13	
CW09B	443,897,471	44,114,780	10.06	
CW12B	556,007,008	52,065,545	10.68	

Figure 2 shows the multiple comparisons after Friedman’s test for NoStem, selective method, and baseline stemmers. It tests column effects (i.e., stemming algorithms, selective method, and NoStem) after adjusting for possible row effects (i.e., queries). The test is appropriate for the multiple comparisons of results of IR experiments since the compared methods are under study and queries do not have any interaction with each other. Tukey’s honestly significant difference criterion (Tukey’s HSD) is employed for multiple comparisons. The figure only shows CW09B and CW12B collections and the remaining collections are given in appendix Section ‘Multiple comparisons’. For CW09B, it appears that the selective method is statistically different from all its base stemmers, but we could not observe the case for other query sets. The selective method with Porter in several cases, such as GOV2 and WSJ, is significantly different from NoStem. Another important point is that the selective method is significantly no worse than the baselines in the experiments, which consisted of the combinations of eight query sets and three stemming algorithms.

Figure 2 Multiple comparisons based on performance scores of the stemmers in IR systems for CW09B and CW12B.

The classifier accuracy of each collection is listed in Table 5. The table contains the correctly predicted and actual number of queries for NoStem and each stemming algorithm. The number of queries with the same performance scores for both NoStem and corresponding stemming is in the Tie column. It is inquired in RQ1 to what degree the proposed selective approach is accurate in predicting the queries that stemming should (not) be applied. The proposed classifier for selective stemming produces moderate accuracy scores for predicting whether or not stemming should be applied. However, the accuracy results only show the classifier performance by ignoring the differences between retrieval scores of the selections. Therefore, the average retrieval system performance score and classifier accuracy needs to be considered together. Thus, Table 2, plots in risk-sensitive analysis (Fig. 3), and plots in per query performance analysis (Figs. 4, 5) show the positive effects of this classifier on the average performance of the retrieval systems by means of the performance evaluation metric for CW09B and CW12B query sets. In addition, the importance of those features in selective stemming is discussed in Section ‘Feature analysis’ by presenting in Fig. 6.

Table 5 The classifier accuracy of the selective approach is presented for each stemmer separately.

Collection	True predicted	Actual	Tie	Accuracy (%)	
	NoStem	KStem	NoStem	KStem			
CW09B	50	17	64	37	96	66	
CW12B	20	33	38	45	102	64	
NTCIR	28	49	60	67	51	61	
GOV2	19	56	52	75	22	59	
WSJ	20	47	49	72	27	55	
MQ07	223	165	366	343	811	55	
MQ08	91	61	154	132	276	53	
MQ09	89	67	149	136	277	55	
Collection	True predicted	Actual	Tie	Accuracy (%)	
	NoStem	Lovins	NoStem	Lovins			
CW09B	72	12	83	45	69	66	
CW12B	21	34	53	52	80	52	
NTCIR	62	19	87	60	31	55	
GOV2	15	50	51	74	24	52	
WSJ	4	43	42	60	46	46	
MQ07	250	182	444	385	691	52	
MQ08	156	38	201	136	225	58	
MQ09	113	55	189	145	228	50	
Collection	True predicted	Actual	Tie	Accuracy (%)	
	NoStem	Porter	NoStem	Porter			
CW09B	59	13	75	48	74	59	
CW12B	22	38	47	57	81	58	
NTCIR	40	44	75	71	32	58	
GOV2	16	71	51	86	12	64	
WSJ	19	59	50	80	18	60	
MQ07	185	244	392	430	698	52	
MQ08	96	71	176	162	224	49	
MQ09	97	78	181	155	226	52	

Figure 3 (A–B) TRisk comparisons of proposed method and baseline stemmers against NoStem for α parameter from 0 to 5.

Figure 4 Per query performance differences for CW09B.

Figure 5 Per query performance differences for CW12B.

Figure 6 (A–B) Feature importance according to the ablation study.

We have attached the figures of the remaining query sets to the Section ‘Appendix’ to keep the reading flow. The plots for the statistical test are given in Fig. 7. The related plots to the risk-sensitive and per query performance analysis are presented in Figs. 8 and 9, respectively. Finally, feature importance plots are presented in Fig. 10.

Risk-sensitive analysis

In a risk-sensitive evaluation for the robustness of an IR system, the term risk refers to the risk of retrieval effectiveness of an IR system for a given particular query worse than the effectiveness of a baseline system for that query; otherwise, it refers to reward. A control parameter α > 0 penalizes the system with respect to the risk-reward trade-off giving more weight to the risk, and where α = 0 refers to the risk and reward having equal weight. To measure the robustness of the selective approach, we utilize TRisk (Dinçer, Macdonald & Ounis, 2014) risk-sensitive evaluation measure. TRisk is a measure based on hypothesis testing with the identification of queries that commit a significance level of risk. This risk measurement makes use of the linear transformation of t statistic used in the Student’s t-test. The threshold values for TRisk are −2 and +2. An IR system with TRisk <  − 2 at α = 0 is under a risk. For an IR system with TRisk > 0, the system is counted in favor of reward. The performance of the system is considered statistically significantly better than the baseline if the values TRisk > 2.

RQ2 inquires the contribution of the proposed selective approach with respect to robustness in retrieval effectiveness of the IR systems that employ stemming. In this respect, Fig. 3 shows the robustness of the proposed selective approach against single systems in TRisk risk-sensitive evaluation metric. In the figure, we have presented CW09B and CW12B results together and the remaining query sets are appended to the Appendix. The plots in the figures compare the robustness of selective method and its baseline stemmers against NoStem. The proposed method is more robust than the system in which the baseline stemming algorithm is applied. This improvement in robustness has been achieved for almost all experiments. For the increasing values of α, we observe that selective method maintains the robustness for the query sets. We observe that the minimum TRisk value among the query sets is −10.03 for the KStem baseline, but it is −4.98 for selective method. The similar results are valid for Lovins and Porter baselines. The selective method has a minimum TRisk value of −7.82, while the baseline Lovins has a minimum value of −12.61. Furthermore, similar results can be obtained for the baseline Porter: the selective method and the baseline have minimum values of −6.78 and −10.30 respectively. Consequently, the proposed method has minimized the risk of failure in IR system, and those results indicate the contribution of this study.

Per query performance analysis

Per query performance analysis is performed to reveal the number of queries that benefit from or are hurt by the proposed method. Figures 4 and 5 show the nDCG@20 score differences on query basis for CW09B and CW12B query sets, sorted in ascending order. The remaining figures of query sets are appended to the Appendix. Each subfigure has three plots: each of them is produced for NoStem vs. Selective(Sel), and NoStem vs. each base stemmer is placed together in corresponding plot. For instance, in the first plot in Fig. 4, performance differences between NoStem and KStem, and between NoStem and SelKStem are plotted using line graph. Similarly, the second and third plots are for Lovins and Porter, respectively. The axis-y in the plot represents the performance difference for each queries. The high values in axis-y indicate a high-performance difference between systems in terms of retrieval effectiveness.

The score of the retrieval effectiveness of each query in the proposed method is one of the scores produced by the participating systems. This is the reason for the increase in the tie queries (performance differences are equal to 0), which is the middle part of the graphics. According to the plots, it is seen that the selective method decreases the number of queries that are negatively affected by the stemming. In Fig. 4, a 25 percent decrease in the amount of negatively affected queries by stemming for KStem is shown when selective stemming is applied. This is also observed in 36 percent and 30 percent for Lovins and Porter respectively. The percentages for Fig. 5 are an 11 percent decrease for KStem and Porter and a 12 percent decrease for Lovins. In addition, the reason for the average performance improvement with the proposed method is not due to successful predictions in a few queries but to successful predictions across queries in general. For instance, a 0.0117 improvement in performance effectiveness is produced by selective stemming against KStem; at the same time, the percentage of negatively affected queries reduces to 25 percent for CW09B. Moreover, the method alleviates a deterioration in performance by accurately predicting as possible the queries placed in the most left part of the plots. The reduction of the areas in the lower left part of the plots reveals minimizing the risk of failure, and it also confirms the implications of the risk-sensitive analysis.

The results of proposed selective method are slightly worse for query sets of Million Query Tracks as listed in Table 2 than the IR system that Porter is applied. Part of the reason of this situation is to not accurately make prediction for a few queries where performance differences between stemming and no-stemming are the largest. For example, the Porter plots in Fig 9F shows that selective method could not accurately predict those a few queries. This error in prediction is most likely due to machine learning method since the machine learning methods only learn from given sample of data. Thereby, the average performance of the system is affected by this situation. However, this does not harm the robustness of the proposed method, but rather maintains the robustness by minimizing the risk of failure across queries.

Feature analysis

Feature analysis investigates to what degree each feature contributes to performance. Our proposed work takes advantage of several features used in binary classifiers for selective stemming. The features have been used in 24 experiments (a combination of eight query sets and three stemming algorithms), so in this section, we have examined the impact of the features on IR effectiveness. Here, the investigation aims to discover the most and the least important features and rank them in terms of the contribution to the performance improvement of the proposed work.

In conducting feature analysis experiments, we have employed an ablation study. This evaluation ranks the features according to their importance. The features are sequentially removed from the feature set. For each iteration, the classifier model is trained, and the average performance of the IR system is obtained. Finally, the performances of the IR system obtained by each reduced feature set are the importance of the corresponding removed feature. When the features are sorted in ascending order according to their importance, the first feature is the most important feature. Because the classifier model trained by the feature set without this feature produces the lowest performance score. Figure 6 shows the plots for CW09B and CW12B query sets and each stemming algorithm. The plots for other query sets are given in Appendix. Feature labels are on the y-axis, and their importance scores are given on the x-axis in the plots. Also, the features are presented in ascending order according to their importance, but the last row in each plot (label IncludeAll) stands for the performance score obtained by the complete feature set. Each feature provides a different degree of contribution to the experiments. For instance, while feature MaxIDF is the most important feature for CW09B with KStem, it is one of the least important features for Porter, and also removing it improves the performance of the IR system more than the IR system using the complete feature set. Table 6 summarizes the plots according to the importance of the features. The table includes the average and median ranking of each feature in experiments. We can make several inferences from the table. In terms of most important and least important features, MaxIDF and AvgIncDF are the candidate features respectively. The average ranks of the features are around the fifth rank out of ten features. If we had a feature around the second rank on average, we could conclude that it is the core feature. Similarly, if we had a feature around the ninth rank on average, we could conclude that it can be removed from the feature set or it is the least important feature. However, according to our conducting experiments, the most important implication is that the features are of almost equal importance since they are around the fifth rank. The table supports that the features provide the robustness and performance improvements of the selective IR systems. In addition, removing some features from the feature set improves the retrieval scores as seen in Fig. 6. Those features are different for each experiments. For instance, removing MaxIDF from the feature set for the experiment conducted on the KStem baseline decreases the performance, but it improves for Porter baseline on CW09B. This is a typical situation because each stemming algorithm changes the term and document frequencies of the terms in a corpus to a different degree. Hence, the terms have different frequency distributions on the corpus. The frequency fluctuations affect the features and their importance in each experiment because machine learning methods learn from data. Hence, each corpus and its terms are individual according to the applied stemming algorithm. The effect may be positive or negative for retrieval performance. However, when all the features are generalized to the experiments, it is seen that each feature contributes to the selective approach.

Table 6 The average and median of the ranks in the feature importance experiments are presented.

Feature	Average	Median	
MaxIDF	4.6	4	
MeanSCQ	4.6	4	
CorrIctfRank	5.2	5	
Chi2-DF-TF	5.5	6	
Mst-Lst-Change	5.5	6	
ModifiedSCS	5.6	6	
Omega	5.7	6	
MaxWeightedIncDF	5.8	6	
Gamma	6	6	
AvgIncDF	6.5	6	

Implications

Theoretical implications

In a selective stemming approach, predicting the selection is solely not enough for evaluating IR system performance. To robust IR system, accurately predicting the selections for the given queries that occur high performance differences between participating systems is crucial. Even if the classifier accuracy is moderate, accurately predicting such queries improve retrieval robustness and performance. Binary classifier has a potential to alleviate the per query performance degradation problem by using the features obtained with term frequency distributions of participating systems. The used classifier appears to achieve this aim by determining useful neighbor queries in decision process. Thereby, the selective approach to stemming minimizes the risk of failure caused by applying stemming to an IR system, and also, according to the experimental results, in most cases, the selective method systematically improves the effectiveness of the systems participating in selection. In most cases, the selective method also systematically improves the effectiveness of the systems participating in selection according to the experimental results. In addition, the proposed work enriches existing literature on selective stemming in terms of improving the robustness and performance of an IR system by predicting to apply stemming to the given query. Also, various features are derived from differences in the term frequency distributions of the systems participating in the selection for this purpose. The contribution of those features to selective stemming is empirically presented.

Practical implications

The proposed selective approach to stemming is convenient to employ an IR system. Because the method works on a document index in which stemming is not applied, and stemming is performed at query time by supplying a conflation set of morphological variants. For instance, document frequencies of the term to be stemmed can be calculated by the posting lists of its morphological variants. Thereby, the classifier features can be calculated at index time or offline, and selective stemming on a query basis is possible to employ in an IR system. Especially for the large-scale Web collections (CW09B and CW12B), the method is reasonably robust against performance fluctuation across queries. The proposed method achieves satisfactory selection, improves the robustness, and, in most cases, acquires a higher average retrieval performance than the single systems.

Conclusions

We proposed a method for the selective application of stemming on a per query basis in order to alleviate the robustness and effectiveness deteriorations caused by the queries that are harmed by stemming. The proposed selective approach to stemming was a binary classifier to decide whether stemming should be applied to a given query. The classifier used a rule-based stemmer and pre-retrieval query performance prediction features gathered from the literature. In addition the features in literature, a set of features derived from the frequency distributions of query terms with and without stemming applied is used. The features used in the experiments play a role in different levels of improving retrieval effectiveness according to the query sets. However, on average, features contribute to the retrieval effectiveness of selective stemming in almost equal proportion. Experimental results show that our selective approach is successful at avoiding the risk posed by the queries that are adversely affected by stemming. It is obtained by accurately predicting the application of stemming to the given query. Furthermore, our selective approach is more effective than a single system in which stemming is systematically applied to all queries. This suggests that our proposed selective approach to stemming is both robust and effective. In our following research, we will focus on the classifiers and features to improve the results of this work. We plan to investigate and derive the features that can discriminate queries affected by stemming. Also, we will test advanced machine-learning techniques and pre-trained language models for this purpose.

Supplemental Information

Supplemental Information 1 Feature values and performance scores of datasets

Each data contains feature values and performance scores of the stemmers for the queries

Click here for additional data file.

Appendix

Multiple Comparisons

See Fig. 7.

Figure 7 Multiple comparisons based on performance scores of the stemmers in IR systems for NTCIR, GOV2, WSJ, MQ07, MQ08, and MQ09.

Robustness

See Fig. 8.

a Figure 8 Continued.

a Figure 8 (A–F) TRisk comparisons of proposed method and baseline stemmers against NoStem for α parameter from 0 to 5.

Per Query Performance Differences

See Fig. 9.

a Figure 9 Continued.

b Figure 9 (A–F) Per query performance differences.

Feature Importance

See Fig. 10.

a Figure 10 Continued.

b Figure 10 (A–F) Feature importance according to the ablation study.

Additional Information and Declarations

Competing Interests

Author Contributions

Data Availability

The authors declare there are no competing interests.

Gökhan Göksel conceived and designed the experiments, performed the experiments, analyzed the data, performed the computation work, prepared figures and/or tables, authored or reviewed drafts of the article, and approved the final draft.

Ahmet Arslan conceived and designed the experiments, performed the experiments, analyzed the data, performed the computation work, authored or reviewed drafts of the article, and approved the final draft.

Bekir Taner Dinçer conceived and designed the experiments, analyzed the data, authored or reviewed drafts of the article, and approved the final draft.

The following information was supplied regarding data availability:

The codes are available at Zenodo:

Ahmet Arslan, gokhanc90, & ahmetalkilinc. (2022). gokhanc90/lucene-clueweb-retrieval-selectivestemming: Selective Stemming Lucene init (v0.2.1-alpha). Zenodo. https://doi.org/10.5281/zenodo.7068020

Gökhan Göksel. (2022). gokhanc90/matlabIRexperiments: Selective Stemming Matlab (v0.2.3-alpha). Zenodo. https://doi.org/10.5281/zenodo.7083370.

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
