# Peer review of "A selective approach to stemming for minimizing the risk of failure in information retrieval systems"

_PeerJ Computer Science, doi:10.7717/peerj-cs.1175_

## Round 0.1 · original submission · Major Revisions

· Academic Editor

Major Revisions

Please revise your paper according to the reviewers' comments.

Thank you very much.

·

Basic reporting

Overall the manuscript is well organized, However there are several grammatical issues as well as inappropriate content as followed.

1 - Page 6, Line 73, "the following section......"; which following section?

2 - The page 7, Figure 1 caption needs to corrections. Some words are not correctly rendered.

3 - Page 9, Line 218, "This query expansion technique is examined....." please use correct the way of discussing published. It should be in past tense.

4 - Page 11: the method to referring to equation is not appropriate. Please use the equation number.

5 - Page 18: section don't provides the in text quantitative descriptions of results such as difference in performance which makes it hard for reader to keep focus. Moreover the presentation of graphs is not appropriate, it is better to make one graph (one for CW09B and one for CW12B) and projecting each case with line graph.

6 - Page 19: Feature analysis section starts with a figure?

7 - page 20: line 545 to 547, In this work......" i think the authors have used this many times. it is better to write in context of implications rather than reintroducing the paper.

8 - Page 21, Conclusion has been written in abstract form which is not a good practice. Conclusion should summarize the new findings in context research questions and their significance and implications.

Experimental design

the experimental approach is well designed.

Validity of the findings

Although authors have all analyses all the relevant aspects of the selective query, However following issue throughout the manuscript.

5 - Page 18: section don't provides the in text quantitative descriptions of results such as difference in performance which makes it hard for reader to keep focus. Moreover the presentation of graphs is not appropriate, it is better to make one graph (one for CW09B and one for CW12B) and projecting each case with line graph.

The same is the case with other sections.

Additional comments

The article needs though review in in terms of presentation of the findings and grammatical correctness.

Reviewer 2 ·

Basic reporting

The paper "A selective approach to stemming for minimizing the risk of failure in information retrieval systems" represents a very good study. The authors are very familiar with the fields covered in the paper. It is well written with all the necessary elements.
The paper has great potential and can be accepted after major corrections:
- Citing a large number of papers in one sentence without any explanation is not popular. Three references are maximum that can be cited in one sentence. For example, in the introduction, you have "It has been a long-standing debate whether stemming contributes to the effectiveness of an IR system (Harman, 1991; Krovetz, 1993; Hull, 1996; Flores and Moreira, 2016; Alotaibi and Gupta, 2018)".
- A lot of old references are cited in the paper. I think that should be removed each old reference which isn't core for the paper.
- Titles of some Figures and Tables are too long. Please shorten it.

Experimental design

- Clearly described aims, the main contributions, novelty, and verification of results should be more concise described in the abstract.
- Please clearly separate the advantages of your paper in more detail.
- What new brings your paper? Novelty and contributions should be well described.

Validity of the findings

- Future research in conclusion is poor, you should add more guidelines.

---

## Round 0.2 · accepted · Accept

· Academic Editor

Accept

Dear Authors,

I am pleased to inform you that your paper has now been accepted for publication. Thank you very much for choosing PeerJ Computer Science.

·

Basic reporting

The authors have addressed all issues and article reporting is fine now.

Experimental design

Authors have addressed all issues identified.

Validity of the findings

All necessary changes have been made by authors in the revised article.

Additional comments

The article is fine and ready for publishing.

Reviewer 2 ·

Basic reporting

The authors have addressed my comments in the proper way.

Experimental design

The authors have addressed my comments in the proper way.

Validity of the findings

The authors have addressed my comments in the proper way.

Additional comments

The authors have addressed my comments in the proper way.